# Root Influences Rhizosphere Hydraulic Properties through Soil Organic Carbon and Microbial Activity

**DOI:** 10.3390/plants13141981

**Published:** 2024-07-19

**Authors:** Aline Martineli Batista, Thaís Nascimento Pessoa, Fernando Ferrari Putti, Fernando Dini Andreote, Paulo Leonel Libardi

**Affiliations:** 1School of Agriculture, São Paulo State University (UNESP), Rua José Barbosa de Barros 1870, Botucatu 18610-307, SP, Brazil; fernando.putti@unesp.br; 2Department of Soil Science, Luiz de Queiroz College of Agriculture (ESALQ), University of São Paulo (USP), Piracicaba 13418-900, SP, Brazil; thaisnpessoa@usp.br (T.N.P.); fdandreo@usp.br (F.D.A.); pllibard@usp.br (P.L.L.); 3School of Sciences and Engineering, São Paulo State University (UNESP), Rua Domingos da Costa Lopes 780, Tupã 17602-496, SP, Brazil

**Keywords:** β-glucosidase, annual ryegrass, dehydrogenase, easily extractable glomalin-related soil protein, hydraulic conductivity, *Lolium multiflorum*, microbial biomass carbon, soil aggregate stability, soil water retention curve

## Abstract

Processes of water retention and movement and the hydraulic conductivity are altered in the rhizosphere. The aim of this study was to investigate the physical-hydric properties of soil aggregates in the rhizosphere of annual ryegrass (*Lolium multiflorum*) cropped in a Kandiudalfic Eutrudox, taking into account aspects related to soil aggregate stability. Soil aggregates from rhizosphere soil (RZS) and soil between plant rows (SBP) were used to determine soil water retention curves (SWRCs) and saturated hydraulic conductivity (*K_sat_*). In addition, properties related to soil aggregate stability, such as water-dispersible clay, soil organic carbon (SOC), and microbial activity, were also assessed. The higher microbial activity observed in the RZS was facilitated by increased SOC and microbial activity, resulting in improved soil aggregation (less water-dispersible clay). For nearly all measured matric potentials, RZS had a higher water content than SBP. This was attributed to the stability of aggregates, increase in SOC content, and the root exudates, which improved soil water retention. The increase in total porosity in RZS was associated with improved soil aggregation, which prevents deterioration of the soil pore space and results in higher *K_sat_* and hydraulic conductivity as a function of the effective relative saturation in RZS compared to SBP.

## 1. Introduction

The rhizosphere soil is the zone of soil surrounding a living root that is influenced by root activity and represents a complex plant–soil interface [1]. The dynamic nature of the rhizosphere creates biophysical and chemical gradients that differ significantly from those observed in the bulk soil [2,3,4]. Because of these differences, the rhizosphere soil has been the subject of several studies [5,6,7,8,9]. However, the physical-hydric processes mediated by the roots in the rhizosphere have received limited consideration, especially in tropical soils.

Previous studies have demonstrated that structural changes in the rhizosphere soil can occur as a result of the rearrangement of its particles, which can modify its aggregates and affect the water uptake and dynamics, gas exchange, and water content in the rhizosphere [4,7,10,11]. Thus, processes, such as water retention and movement, and the hydraulic conductivity property must also be altered in the rhizosphere. Understanding soil hydraulic properties is critical for assessing environmental impacts in agricultural systems and for better planning and management of water resources [12,13].

A key challenge for agriculture in the 21st century is to optimize the use of water resources [5]. Since water and nutrients must pass through the rhizosphere before being taken up by roots, and since approximately 40% of terrestrial precipitation passes through the rhizosphere [2,14], managing the physical-hydric properties of the rhizosphere to enhance plant resilience to various abiotic stresses may be a solution [5,15]. To do this, it is essential to know and understand the hydraulic properties of the soil in the rhizosphere.

Some models demonstrate the influence of the mucilage (i.e., a polymeric gel released by most plant roots) on the hydraulic properties of the rhizosphere soil [9,16,17,18]. However, in order to develop these models, researchers often use mucilage derived from chia (*Salvia hispanica*) seeds as a proxy for natural mucilage in sandy soils. Apparently, the presence of mucilage in soil enhances the water retention capacity and reduces the evaporative flux [7,17,19]. Additionally, it has been observed that mucilage decreases saturated hydraulic conductivity (*K_sat_*) while increasing unsaturated hydraulic conductivity (*K*) [7,17,19,20,21]. During drying and rewetting cycles, mucilage induces a distinct hysteretic behavior. Specifically, the rhizosphere remains moist for a longer period during drying and rehydrates more slowly than the bulk soil during rewetting [7,17,18,22,23].

A previous study evaluated the soil water retention in aggregates from the rhizosphere soil of maize (*Zea mays*), wheat (*Triticum aestivum*), and barley (*Hordeum vulgare*) cropped in a silty clay loam soil and reported lower water retention for aggregates from the rhizosphere soil of barley than from the bulk soil [10]. They believed that changes in the contact angle between water and soil and surface tension could influence changes in water retention behavior. Nevertheless, they stressed the need for further studies to validate the effect of root activity on the contact angle. To the best of our knowledge, currently no study has reported the effects of roots and their rhizosphere on the physical-hydric properties of a tropical soil.

Although particular attention has been given to the mucilage—recognized as the main driver of changes in the physical-hydric properties of the rhizosphere soil—the exudation of all organic compounds from the roots (i.e., root exudates) can induce changes in these properties [6,24,25,26]. This exudation is associated with an increase in carbon deposition and soil aggregation in the rhizosphere [27]. However, it depends on several factors, including soil (e.g., soil structure, soil texture, pH, and presence of microorganisms) and plant (e.g., plant species and age) characteristics [2,27,28].

Root exudates are composed of amino acids and amides, phenolic acids, coumarins, enzymes, sugars, organic acids, and others [29]. On the one hand, organic acids, which are present in large amounts in some root exudates, may weaken the soil particles [28,30]. On the other hand, sugars that include polysaccharides may contribute to the stabilization of soil particles [28]. An increase in the carbon content of the rhizosphere soil of a consortium of black oats (*Avena strigosa*) and vetch (*Vicia sativa*), which did not result in an increase in aggregate stability was reported in an Oxisol [27]. This was likely due to the high concentration of organic acids in the root exudates produced by vetch.

The observation of structural differences may be more challenging in highly stable soils, such as those found in tropical regions, which typically have high levels of exchangeable aluminum and iron oxides due to the high level of weathering, providing high structural stability. The formation of well-developed and stable microstructure in these soils is a consequence of the distribution of surface charges on kaolinite minerals and their coexistence with oxides [31]. Nevertheless, a previous study stressed changes in soil aggregates from the rhizosphere of annual ryegrass (*Lolium multiflorum*) in a tropical soil with a high degree of weathering (Rhodic Hapludox) due to an increase in organic carbon in the rhizosphere soil [27]. Therefore, this crop is likely an effective model for investigating the influence of roots and their rhizosphere.

The application of in situ methods to determine soil hydraulic properties remains challenging due to the limitations of the methodological approach, costs, and the time-consuming nature of conducting experiments, especially in the rhizosphere soil [8,32]. Since the rhizosphere is a few millimeters around the root, obtaining direct measurements of the physical-hydric properties is very difficult, even using modern technologies [15,33]. Consequently, indirect methods, which primarily employ the soil–water retention curve approach and rely on the application of several empirical equations [34,35,36,37], can prove to be highly informative in elucidating intricate interactions among the root system, soil, and hydrological environment.

The hypothesis of this study is that:*(i)*.The aggregates from the rhizosphere (RZS) of annual ryegrass present higher organic carbon content and microbial activity than aggregates from the soil between plant rows (SBP) (i.e., bulk soil);*(ii)*.Organic carbon input and microbial activity lead to higher aggregate stability in the RZS than in SBP;*(iii)*.Organic carbon input, microbial activity and aggregate stability in the RZS increase the RZS porosity, particularly micropores;*(iv)*.The increase in micropores and aggregate stability, supported by the effect of organic carbon input and microbial activity, enhances the water retention capacity in the RZS;*(v)*.Due to the residual effect of the mucilage, *K_sat_* is lower in the RZS than in the SBP;*(vi)*.*K* reflects soil quality and is higher in the RZS than in the SBP.

Thus, the objective of this study is to investigate the physical-hydric properties in the soil aggregates of the annual ryegrass rhizosphere, taking into account soil aspects related to the soil aggregate stability such as water-dispersible clay, soil organic carbon (SOC), and microbial activity. To achieve this objective:*(i)*.The SOC content and microbial parameters [microbial biomass carbon (MBC), β-glucosidase (BG), dehydrogenase (DH), easily extractable glomalin-related soil protein (EE-GRSP)] were determined;*(ii)*.The aggregate stability was assessed by readily dispersible clay (RDC);*(iii)*.The frequency and percentage of pores were obtained from the soil water retention curves (SWRCs);*(iv)*.The water retention capacity was also obtained from the SWRCs;*(v)*.*K_sat_* was determined by a constant head permeameter method;*(vi)*.*K* was derived from the water retention characteristics, with *K_sat_* as the relative hydraulic conductivity, (*K_r_*) as a function of the effective relative saturation (*ω*), and *K* as a function of the *ω*.

## 2. Results

### 2.1. Soil Parameters Related to Soil Aggregation

Soil physical [readily dispersible clay (RDC)], microbiological [microbial biomass carbon (MBC), β-glucosidase (BG), dehydrogenase (DH), easily extractable glomalin-related soil protein (EE-GRSP)], and chemical (SOC) parameters showed superior soil conditions for RZS compared to SBP (Table 1), indicating the positive effect of annual ryegrass on the soil. Only DH activity did not present differences between RZS and SBP. This parameter was not found to be correlated with other microbiological parameters (MBC and EE-GRSP) (Figure 1). All microbiological parameters showed significant positive correlations with SOC (r = 0.68 to 0.93) and significant negative correlations with RDC (r = −0.52 to −0.90). Additionally, the SOC was also negatively correlated with RDC (r = −0.86).

### 2.2. Soil Hydraulic Properties

The SWRC parameters are presented in Table 2. The values of the coefficient of determination (R²) (greater than 0.99) and of the root mean square error (RMSE) (less than 0.01) indicate a good fit between observed and fitted data by the van Genuchten model, for both RZS and SBP environments. Considering the 95% confidence interval, the SWRCs differed between RZS and SBP. For the majority of the measured matric potentials (*Ψ_m_*), the RZS had higher volumetric soil water content (*θ*) than the SBP (Figure 2). The results showed that *θ* corresponding to the field capacity (*θ_FC_*) and to the permanent wilting point (*θ_PWP_*) were higher (*p* < 0.05) in the RZS than in the SBP. However, no significant differences in the available water contents (*θ_AWC_*) were observed between treatments (Figure 3a).

The value of the most frequent equivalent radius (*r_max_*) for both RZS and SBP were in the mesopore class, with a higher size in the RZS (40.98 µm) than in the SBP (32.69 µm) (Figure 4). The class with a higher number of pores was the micropore class for both RZS and SBP (Figure 3b). RZS had a higher (*p* < 0.05) porosity (0.584 ± 0.006 m^3^ m^−3^) than SPB (0.510 ± 0.006 m^3^ m^−3^), with higher porosity in all the classes (Figure 3b).

Saturated hydraulic conductivity (*K_sat_*) was higher (*p* < 0.05) for SRZ (10.34 ± 1.80 mm h^−1^) than for SBP (4.46 ± 0.60 mm h^−1^), but the relative hydraulic conductivity (*K_r_*) as a function of the effective relative saturation (*ω*) [*K_r_*(*ω*)] curves showed no difference between RZS and SBP (Figure 5a). On the other hand, considering the 95% confidence interval, the hydraulic conductivity (*K*) as a function of *ω* [*K*(*ω*)] showed higher values for RZS than for SBP (Figure 5b).

## 3. Discussion

The RZS showed the highest (*p* < 0.05) microbial activity and mycorrhizal fungi (BG, EE-GRSP) and the lowest RDC (i.e., the highest soil aggregation) (Table 1). The improvement of microbial activity was mediated by the highest amount of soil organic carbon (SOC) and microorganisms (MBC) in the RZS, which was also reflected in the improvement of soil aggregation in the RZS (i.e., lowest RDC) (Figure 1). The same observations for natural ecosystems were also observed for ryegrass crop: the RZS provides a favorable environment for microbial activities due to the increased aggregate stability and nutrient input, leading to an increase in enzyme activity and microbial biomass, thus promoting soil carbon mineralization [38,39].

A higher RDC in the SPB than in the RZS (Figure 1) indicates the lowest stability and structural quality of this environment. Soil aggregation is highly correlated with clay dispersion, as flocculation of clay particles is a prerequisite for the formation of water-stable aggregates [40,41,42]. In turn, aggregate stability is modified by the root system as demonstrated in previous works [27,43].

The cultivation of annual ryegrass is largely associated with improvements in soil aggregation, which can be attributed to the architecture of the root system (high number of fine roots), the high population of mycorrhizal hyphae, and the high exudation of polysaccharides, which are known to contribute to the stabilization of soil particles [28,44,45,46]. Additionally, plants can exude other organic molecules from their roots, which act as binding agents of soil particles [6,24,25].

The highest SOC content in the SRZ (Table 1) is related to the exudates released by the roots, which represent the main carbon input to the rhizosphere, or even to the bioavailability of the mineral-associated organic matter that can be mobilized and solubilized by low molecular weight root exudates [45,47,48,49]. The increase in SOC in the SRZ (Table 1) was reflected by the highest microbial activity in this environment, as shown by the positive correlations between SOC and microbiological parameters (BG, DH, MBC, and EE-GRSP) (Figure 1). On the other hand, the increase in microbial activity also reflects an increase in SOC due to decomposition processes mediated by microorganisms, as demonstrated in previous works [38,39].

The increase in SOC was positively correlated with the increase in the BG and DH activity (Figure 1), as these are soil enzymes related to the carbon cycle. BG is involved in the enzymatic degradation of cellulose, while DH is involved in microbial oxidation [50,51]. However, DH activity was the only parameter that did not differ between RZS and SPB (Table 1). Our evaluation was carried out 176 days after sowing, a period when DH activity cannot be stressed in the RZS. A previous study showed that the increase in DH in the RZS occurred 75 days after sowing for different plant species due to the higher root exudation observed during this period [52].

Compounds released by the roots are readily utilized by rhizosphere microorganisms to facilitate essential processes such as growth and respiration [3,53]. Consequently, roots regulate the microbial community in the rhizosphere through the exudation of various compounds [54,55]. This regulation, in turn, enhances the microbial degradation of SOC, which is primarily driven by the production of extracellular enzymes by microorganisms stimulated by the presence of plants [54].

The SOC and all microbiological parameters (BG, DH, MBC, EE-GRSP) were negatively correlated with RDC (Figure 1). The association of plants with microorganisms favors a higher index of aggregate stability in the RZS in response to greater enzymatic activity [56,57,58]. The increase in organic carbon in the rhizosphere, through exudation, leads to an increase in the abundance of decomposing microorganisms, which contributes to greater aggregate stability [59,60].

In addition, the association of roots with fungi also contributes to the soil aggregation process [56,61]. As such, as EE-GRSP is a glycoprotein synthesized mostly by arbuscular mycorrhizal fungi [62], it is able to detect the presence of mycorrhizal hyphae and shows a strong positive relationship with soil aggregation and enzyme activities in the rhizosphere [56,63,64].

Regarding the soil hydraulic properties, for the majority of the measured *Ψ_m_*, the RZS had a higher *θ* than SBP (Figure 2), thereby demonstrating its superior water retention capacity. The greater number of micropores in the RZS than in the SBP (Figure 3b) leads to the highest water retention capacity in these soil aggregates, as micropores are textural pores responsible for increased soil water retention [65]. The higher organic carbon content in this environment (Table 1) also contributes to an increase in the water retention capacity [12,66].

It has been reported that the decomposed soil organic matter (SOM) improves the water retention due to factors such as low bulk density, high porosity, and high specific surface area of these composts [66]. Furthermore, SOM is the main factor influencing the differences in soil water retention under comparable clay content or textural conditions [67], as in the case of this study.

It is also evident that the mucilage exuded by the roots has the capacity to absorb large amounts of water, increasing the water retention capacity of the soil [5,9,18,68]. Even if the assessment was not performed with active roots, the changes promoted by annual ryegrass residues in the soil can remain in the soil for some time [41].

Although disturbed samples were used to determine SWRCs (Figure 2) and consequently to assess *θ_FC_*, *θ_PWP_*, and *θ_AWC_* (Figure 3a) and pore size distribution (Figure 3b), recent and more advanced studies that allow in situ visualization of the root–soil interface through 3D X-ray computed tomography analysis show that total porosity increases in the RZS [58,69,70]. The occurrence of the highest porosity in the RZS is thought to be due to root growth, and increases in both larger and smaller pores have been observed [10,71]. Recent studies also found that root-mediated physical and biological processes could also increase the RZS porosity through enhancing aggregation [4,72].

Therefore, the observed increase in the SRZ porosity may be attributed to an improvement in soil aggregate stability due to the best microbial activity (Table 1), as degradation of soil structural stability can lead to the deterioration of the soil hydraulic network, i.e., the network of soil pores [42]. These results show that the access to physical-hydric properties with classical and indirect methodologies corroborates with the most advanced studies.

The highest *θ* for almost all *Ψ_m_* in the RZS (Figure 2) resulted in higher (*p* < 0.05) *θ_FC_* and *θ_PWP_* in the RZS than in the SBP (Figure 3a). Therefore, *θ_AWC_* did not show differences between treatments (Figure 3a), as the RZS and the SBP did not show differences in *θ* in SWRC between *Ψ_m_* −4.0 to −33.0 kPa (Figure 2). These results do not accurately reflect the actual *θ_AWC_* to the plants, as a study comparing SWRC from disturbed and undisturbed soil samples in an area very close to the study area demonstrated that for *Ψ_m_* less negative than −10 kPa, disturbed soil had a higher *θ* than undisturbed soil [73]. Therefore, advances in technologies are required to provide in situ studies to best estimate the true difference between *θ_AWC_* in the rhizosphere and bulk soil.

The *α*, *m*, and *n* parameters of Equation (2) (Table 2) are related to the pore size distribution and are referred to as shape parameters [74,75]. The *r_max_* was observed to be higher in the RZS (40.98 µm) than in the SBP (32.69 µm). The position of the maximum pore frequency (i.e., the position of *r_max_*) is indicated by the *α* parameter [74]. However, no significant differences (*p* < 0.05) were observed between the *α* and *r_max_* values of the RZS and SBP (Table 2 and Figure 4).

In a previous study, *r_max_* values observed for disturbed soil samples were practically the same for Oxisols of different mineralogical and textural classes [12]. Thus, the difference observed in the present study can be attributed to the root system of the annual ryegrass. The *m* and *n* parameters did not differ between SRZ and SBP (Table 2). These parameters are related to pore width and size [74]. This justifies the identical sequence of porosity for both SRZ and SBP, as well as the *r_max_* within the same class for both of them (Figure 3b and Figure 4).

The higher soil aggregation (i.e., lower RDC) in the SRZ than in the SBP (Table 1) leads to higher *K_sat_* in the SRZ (10.34 ± 1.80 mm h^−1^) than in the SBP (4.46 ± 0.60 mm h^−1^). Determination of the fraction of water-dispersible clay by turbidimetry allows inference of many other physical properties and processes, including soil stability and structural quality and water percolation [76,77]. The effect of aggregate stability was found to be a more significant factor in determining *K_sat_* than the possible presence of mucilage in the RZS. This is because the presence of mucilage is expected to decrease *K_sat_* in the RZS [9,19].

As no differences were found for parameter *m* obtained from Equation (2) (Table 2), *K_r_*(*ω*) curves did not reveal any significant differences between RZS and SBP (Figure 5a). On the other hand, *K*(*ω*) showed higher values for RZS than for SBP (Figure 5b), thereby providing insight into the actual hydraulic conductivity conditions observed in these environments to incorporate the *K_sat_* values. As measuring unsaturated hydraulic conductivity is difficult, *K_r_*(*ω*) and *K*(*ω*) curves derive this function from the water retention characteristics and *K_sat_*. Thus, the superior *K*(*ω*) values observed for RZS can be attributed to the better aggregation and porosity observed for RZS compared to SBP, as evidenced by the lowest RDC value (Table 1) [78,79].

In short, the results presented in this article demonstrate that root-mediated physical (RDC), chemical (SOC) and biological (MBC, BG, DH, and EE-GRSP) parameters increase the porosity through enhancing aggregation and favor soil water retention, *K_sat_* and *K*.

## 4. Materials and Methods

### 4.1. Field Experiment and Soil Sampling

A field experiment was carried out to obtain soil aggregates directly from the rhizosphere of annual ryegrass in an experimental area of 10.8 × 8.0 m^2^ and homogenous elevation and slope located in Piracicaba, São Paulo, Brazil (22°42′15.0″ S 47°37′23.3″ W, altitude 564 m) (Figure 6). The soil of the experimental area was classified as Kandiudalfic Eutrudox [80,81] with a clay texture (456 g clay kg^−1^ soil, 161 g loam kg^−1^ soil and 383 g sand kg^−1^ soil). The climate of the region is tropical with dry winters (Aw), according to the Köppen classification [82].

In the area where the annual ryegrass was grown, lime was applied to the entire area in February 2019 at a rate of 910 kg ha^−1^. In April 2019, the annual ryegrass (cv. BRS Ponteio) was sown with a line spacing of 0.17 m. Two weeks after planting, potassium fertilization was applied with potassium chloride (KCl) at a rate of 80 kg ha^−1^, followed by nitrogen fertilization with urea at a rate of 50 kg ha^−1^. Manual weed control was performed weekly, and in June 2019, the herbicide Heat (70 g ha^−1^) was applied to support weed control.

The experiment was designed as a complete randomized block design, with five replicates and two treatments. The treatments were the rhizosphere soil (RZS) and the soil between plant rows (SBP). At 176 days after sowing, RZS and SBP were collected from the top-soil layer (0.00–0.10 m). Soil adhering to the roots (RZS) was carefully removed from the roots by brushing after gently shaking the roots by hand, and the aggregates were carefully removed by brushing [27].

For readily dispersible clay (RDC), soil organic carbon (SOC) and all microbial parameters, the five replicates were used. However, due to the volume of soil required to determine the hydraulic properties, the samples from the five points were pooled to obtain a composite sample. Three soil samples were then selected for each treatment (RZS and SBP). Air-dried aggregates less than 2 mm were used for all soil determinations described below.

### 4.2. Soil Aspects Related to Soil Aggregation

#### 4.2.1. Readily Dispersible Clay in Water

The readily dispersible clay (RDC) method of turbidimetry was employed to assess soil aggregate stability [83]. This method was adopted in previous studies due to its capacity to detect differences between RZS and SBP [27,84,85].

Aliquots of 5 g of the air-dried soil aggregates and 125 mL of deionized water were added to 150 mL flasks. The flasks were shaken manually (four inversions per minute) and left to rest for 24 h. The suspension was read in a turbidimeter model 2100AN (HACH, Loveland, CO, USA) in NTU (nephelometric turbidity units), with turbidity values being directly proportional to the number of colloids in suspension—in this case, clay. The residual water content in the air-dried aggregates was determined to obtain the mass of dry aggregates. The turbidity was normalized to a concentration of 1 g L^−1^ to account for the effects of varying water content in the air-dried aggregates. The normalized turbidity was expressed in NTU/(g L^−1^), following Equation (1):(1)$$NT=\frac{T}{1000\;\frac{m_{s}}{125}}$$
where *NT* is the normalized turbidity [NTU/(g L^−1^)], *T* is the turbidity (NTU), *m_s_* is the dry soil mass (g); 125 is the volume of water used (mL), and 1000 is a correction factor to transform mL in L.

#### 4.2.2. Microbial Parameters

Microbial biomass carbon (MBC), the activities of two soil enzymes related to the carbon cycle—β-glucosidase (BG) and dehydrogenase (DH)—and the easily extractable glomalin-related soil protein (EE-GRSP), were determined immediately after soil sampling with preserved field water content.

The MBC was extracted by an indirect fumigation–extraction method and determined by titration [86]. The BG activity was determined using 1 g of soil following the method proposed by Tabatabai [87]. The DH activity was determined using 5 g of soil following the method proposed by Casida et al. [88]. EE-GRSP was extracted from the soil according to the method of Wright and Upadhyaya [89] and quantified by the Bradford test [90]. 

#### 4.2.3. Soil Organic Carbon

The soil organic carbon (SOC) was determined by the wet method by oxidation with potassium dichromate [91].

### 4.3. Soil Hydraulic Properties

#### 4.3.1. Soil–Water Retention Curve

To determine soil–water retention curves [SWRCs: volumetric soil water content (*θ*) as a function of matric potential (*Ψ_m_*)], the soil aggregates were conditioned in acrylic cylinders (0.07 m diameter and 0.03 m height) with bulk densities similar to the field conditions.

The samples were gradually saturated with deionized and de-aerated water for 24 h. After saturation, the samples were weighed in order to determine their volumetric water content at saturation (*θ_s_*). The SWRCs were determined in Haine’s funnels (*Ψ_m_* of −0.5, −1.0, −2.0, −4.0, −6.0, and −10 kPa) and in Richards’ pressure chambers (*Ψ_m_* of −30, −100, −500, and −1500 kPa) by quantifying *θ* for each *Ψ_m_* after hydraulic equilibrium was reached [92].

The SWRC data were fitted to Equation (2) proposed by van Genuchten [35], considering Mualem’s restriction, where the parameter *m* = 1 − (1/*n*) [34]:(2)$$\theta =\theta_{r}+\frac{\left(\theta_{s}-\theta_{r}\right)}{{\left[1+{\left(\alpha \;|\Psi_{m}|\right)}^{n}\right]}^{m}}$$
where *θ* is the soil water content (m^3^ m^−3^), *Ψ_m_* is the matric potential of water in the soil (|kPa|), *θ_r_* is the residual soil water content (corresponding to the *Ψ_m_* of −150 m), *θ_s_* is the soil water content at saturation (m^3^ m^−3^), *α* (m^−1^), *m* and *n* are empirical parameters of the model, where *m* and *n* are dimensionless.

The permanent wilting point (*θ_PWP_*) was considered as *θ* corresponding to the *Ψ_m_* of −1500 kPa and the field capacity (*θ_FC_*) as the *θ* corresponding to the *Ψ_m_* of −33 kPa. The available water content (*θ_AWC_*) was calculated as the difference between *θ_FC_* and *θ_PWP_*.

#### 4.3.2. Pore Size Classification and Pore Size Frequency

The pore size classification and the pore size frequency were obtained from SWRCs based on capillarity theory using the Young–Laplace equation [12,75].

The pore size classification was obtained considering micropores (pores with a radii < 15 μm or |*Ψ_m_*| > 10 kPa), mesopores (pores with radii > 15 μm and <50 μm or 3 kPa < |*Ψ_m_*| < 10 kPa), and macropores (pores with a radii > 50 μm, or |*Ψ_m_*| < 3 kPa) [93].

The pore size frequency was obtained by replacing |*Ψ_m_*| in Equation (2) for $\frac{2\;\sigma \;}{\rho_{w}\;gr}$ from the capillarity theory, resulting in Equation (3):(3)$$\theta ={\;\theta }_{r}+\frac{\left(\theta_{s}-\theta_{r}\right)}{{\left[1+{\left(\frac{R}{r}\right)}^{n}\right]}^{m}}$$
where *R* is $\frac{2\;\sigma \;}{\rho_{w}\;g}$; σ is the water surface tension (0.07194 N m^−1^); $\rho_{w}$ is the water density (1000 kg m^−3^); g is the acceleration of gravity (9.8 m s^−2^) and *r* is the equivalent radius (m).

Differentiating Equation (3) with respect to log(*r*) results in Equation (4):(4)$$\frac{d\theta \;}{dlog(r)}=\frac{\theta_{s}-\theta_{r}}{\theta_{s}}\;mn({R)}^{n\;}r^{-n}{[1+({R)}^{n}r^{-n}]}^{-m-1}$$

The value of the most frequent equivalent radius (*r_max_*) was obtained by Equation (5), which is the equation resulting from the differentiation of Equation (4) with respect to log(*r*) equaled to zero:(5)$$r_{max}=\frac{2\;\sigma \;\alpha \;}{\rho_{w}\;g}{\left(\frac{1}{m}\right)}^{-\frac{1}{n}}$$

#### 4.3.3. Hydraulic Conductivity

Soil aggregates were placed in metal cylinders (0.053 m diameter and 0.048 m height) as described for the acrylic cylinders used to determine SWRCs. Saturated hydraulic conductivity (*K_sat_*) was determined using the constant head permeameter method [92,94]. Samples were saturated with deionized and de-aerated water for 24 h. A constant water head was then maintained on the surface of the sample using Mariotte flasks. Once the steady-state condition was reached, the Darcy–Buckingham equation [Equation (6)] was applied to obtain *K_sat_*:(6)$$K_{sat}=\frac{V_{w}\;L\;}{A\;t\;(h+L)}$$
where *V_w_* is the volume of water (mm^3^) collected during time, *t* (h); *A* is the cross-sectional area of the sample (mm^2^), *L* is the length of the sample (mm), *h* is the constant water head at the top of the sample (mm).

To obtain the unsaturated hydraulic conductivity (*K*) from the water retention characteristics and *K_sat_*, the relative hydraulic conductivity (*Kr*) as a function of the effective relative saturation (*ω*) was calculated by Equation (7) resulting from the van Genuchten model [35] based on the Mualem model [34]:(7)$$K_{r}=\omega^{l}{\left[{1-\left({1-\omega }^{\frac{1}{m}}\right)}^{m}\right]}^{2}$$
where *l* is an empirical parameter estimated by Mualem [34] with an approximate generalized value of 0.5 for most soils, *m* is one of the parameters of Equation (1), and *ω* is the effective relative saturation defined as (*θ* − *θ_r_*)/(*θ_s_* − *θ_r_*).

Then, the hydraulic conductivity (*K*) as a function of *ω*, the *K*(*ω*) function, was estimated by the product *K_sat_ × K_r_*(*ω*).

### 4.4. Data Analysis

The data from SWRCs were fitted using RETC software version 1.0 [95]. Comparisons of SWRC, *K_r_(ω)*, and *K(ω)* curves between the treatments (RZS and BPS) were performed using a 95% confidence interval. If there was no overlap between the upper and lower limits of the confidence interval, a significant difference was considered [96]. The effectiveness of the fitted data of SWRCs was evaluated by the coefficient of determination (R^2^) and the root mean square error (RMSE).

The R software version 4.2.3 was used for statistical procedures [97]. To compare the differences between treatments for the other soil properties, the means were compared by *t*-test, with a significance level of 0.05. In addition, a Pearson correlation was used to observe the relationship between them. The results and their interrelationships were used to explain the observed phenomena related to the physical-hydric properties.

## 5. Conclusions

Higher carbon content and microbial activity were observed in the RZS than in the SBP. This is likely due to the high release of polysaccharide-rich exudates, which act as soil-binding agents, as previously described by other authors [28,46]. The improvement in SOC and microbial activity in the RZS increases aggregate stability (i.e., decreases the RDC) and the percentage of pores of all classes in this environment. However, the *r_max_* was not significantly affected. Nevertheless, the increase in micropores, SOC, and aggregate stability leads to higher soil water retention capacity in the RZS than in the SBP. The influence of root exudates probably contributes to the improvement of soil water retention capacity; however, further studies are needed to elucidate their influence in situ. Additionally, the effect of mucilage was not observed in the *K_sat_*, where the effect of soil aggregate structure prevailed, resulting in a higher *K_sat_* in the RZS than in the SBP. *K(ω)* also reflected the aggregation and porosity and stressed the higher soil physical quality in the RZS than in SBP.

The use of disturbed soil sampling and the absence of activity comprise the main limitations of this study. Although the indirect methods considered have provided relevant insights into how roots modulate the soil aggregates in their rhizosphere and their effect on physical-hydric properties, the development of more advanced and economically accessible techniques is needed to investigate in situ differences between RZS and SBP, considering the effect of active roots and their exudates.

## Figures and Tables

**Figure 1 plants-13-01981-f001:**
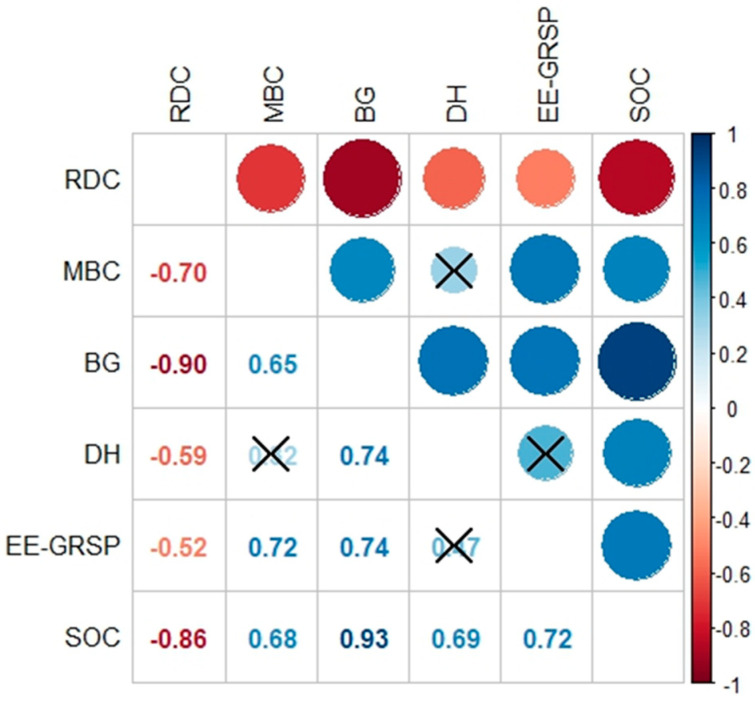
Correlation matrix with the plotted significance test. Negative and positive correlations are shown as red and blue circles, respectively. Size and color intensity of the circles are proportional to the correlation coefficients. The values of the correlation coefficients are shown in the color intensity bar. Crosses were added to correlations that were not considered significant at the 0.95 confidence level. RDC: readily dispersible clay [NTU/(g L^−1^)]; MBC: microbial biomass carbon (mg g^−1^); BG: β-glucosidase (mg PNF kg^−1^ soil h^−1^); DH: dehydrogenase (µg TPF g^−1^ soil 24 h^−1^); EE-GRSP: easily extractable glomalin-related soil protein (mg g^−1^); SOC: soil organic carbon (g kg^−1^).

**Figure 2 plants-13-01981-f002:**
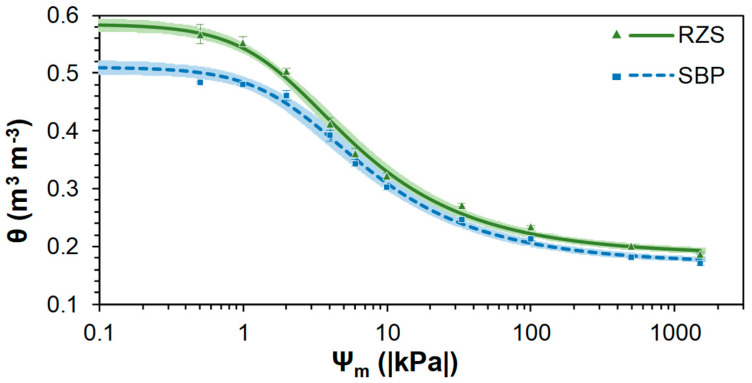
Soil water retention curves (SWRCs) for aggregates from the rhizosphere soil (RZS) and aggregates from the soil between plant rows (SBP). Points represent the mean of the observed data ± standard error, lines correspond to the fitted data, and shadows represent a 95% confidence interval, n = 3.

**Figure 3 plants-13-01981-f003:**
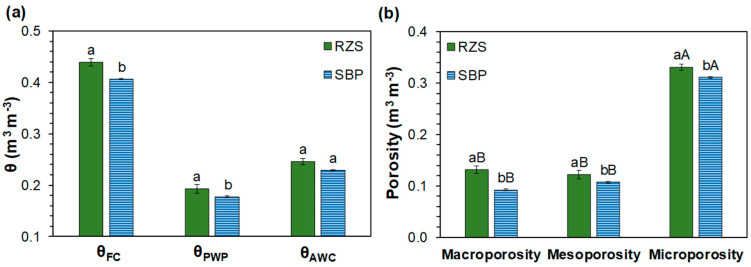
(**a**) Volumetric soil water content (*θ*) corresponding to the field capacity (*θ_FC_*) and to the permanent wilting point (*θ_PWP_*), and the available water contents (*θ_AWC_*) in the aggregates from the rhizosphere soil (RZS) and aggregates from the soil between plant rows (SBP), and (**b**) pore size distribution. Lowercase letters indicate differences between RZS and SBP, and uppercase letters indicate differences between fractions of porosity for each treatment at the 5% significance level by the *t*-test. Data are mean ± standard error, n = 3.

**Figure 4 plants-13-01981-f004:**
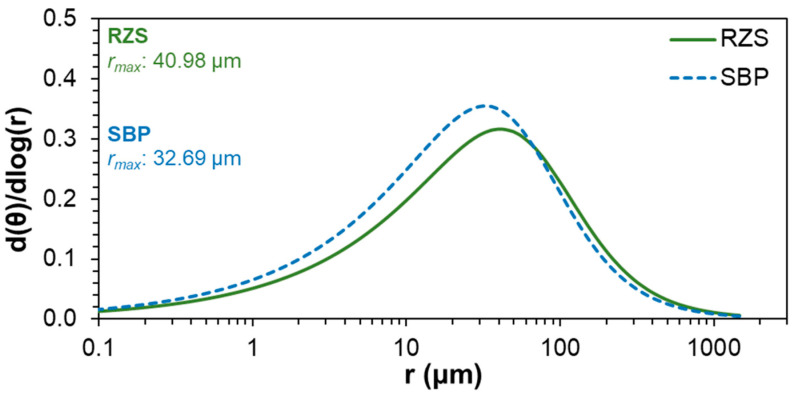
Curves of radius frequency per logarithm of radius as a function of logarithm of radius for aggregates from the rhizosphere soil (RZS) and aggregates from the soil between plant rows (SBP).

**Figure 5 plants-13-01981-f005:**
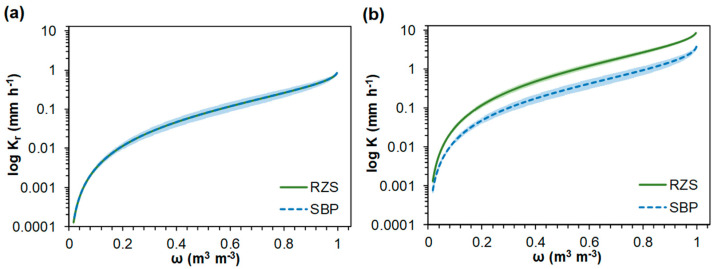
(**a**) Relative hydraulic conductivity (*K_r_*) as a function of the effective relative saturation (*ω*), and (**b**) hydraulic conductivity (*K*) as a function of the effective relative saturation (*ω*) for aggregates from the rhizosphere (RZS) and aggregates from the soil between plants (SBP). Data are mean ± 95% confidence interval, n = 3.

**Figure 6 plants-13-01981-f006:**
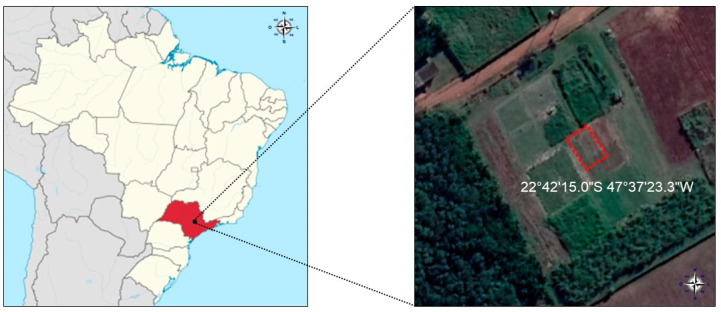
Local development of the field experiment, Piracicaba, São Paulo, Brazil (22°42′15.0″ S 47°37′23.3″ W).

**Table 1 plants-13-01981-t001:** Readily dispersed clay (RDC), microbiological parameters (MBC, BG, DH, EE GRSP), and soil organic carbon (SOC) for rhizosphere soil (RZS) and soil between plants (SBP).

Parameters	Unit	RZS			SPB		
RDC	NTU/(g L^−1^)	0.41	±0.77	b	0.82	±1.49	a
MBC	mg g^−1^	0.98	±0.04	a	0.76	±0.06	b
BG	mg PNF kg^−1^ soil h^−1^	175.50	±7.55	a	100.54	±6.15	b
DH	µg TPF g^−1^ soil 24 h^−1^	3.27	±0.03	a	3.14	±0.05	a
EE-GRSP	mg g^−1^	46.69	±2.07	a	35.79	±3.24	b
SOC	g kg^−1^	42.16	±0.86	a	35.84	±0.60	b

RDC: readily dispersible clay; MBC: microbial biomass carbon; BG: β-glucosidase; DH: dehydrogenase; EE-GRSP: easily extractable glomalin-related soil protein; SOC: soil organic carbon. Letters show differences between aggregates from the rhizosphere (RZS) and aggregates from the soil between plants (SBP) at the 5% level of significance by the *t*-test. Data are mean ± standard error, n = 5.

**Table 2 plants-13-01981-t002:** Parameters obtained from Equation (2) fitted to the soil–water retention curve (SWRC) data for rhizosphere soil (RZS) and soil between plants (SBP).

Parameters	Unit	RZS			SPB		
*θ_s_*	m^3^ m^−3^	0.584	±0.0058	a	0.510	±0.0062	b
*θ_r_*	m^3^ m^−3^	0.186	±0.0023	a	0.171	±0.0015	b
*m*		0.381	±0.0021	a	0.381	±0.0084	a
*n*		1.614	±0.0054	a	1.615	±0.0223	a
*α*	m^−1^	5.081	±0.2827	a	4.107	±0.4106	a
R^2^		0.995	±0.0007		0.995	±0.0007	
RMSE		0.010	±0.0010		0.009	±0.0008	

*θ_s_*: volumetric water content at saturation; *θ_r_*: the residual soil water content; *α*, *m* and *n*: empirical parameters of the model; R^2^: coefficient of determination; RMSE: root mean square error. Letters show differences between aggregates from the rhizosphere (RZS) and aggregates from the soil between plants (SBP) at the 5% level of significance by the *t*-test. Data are mean ± standard error, n = 3.

## Data Availability

The raw data supporting the conclusions of this article will be made available by the authors upon request.

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
