# Peer review of "Root Influences Rhizosphere Hydraulic Properties through Soil Organic Carbon and Microbial Activity"

_plants, 2024, doi:10.3390/plants13141981_

Round 1

Reviewer 1 Report

Comments and Suggestions for Authors

1. The current introduction provides a brief description of the research background. It is recommended to include more information on the significance of rhizosphere soil physical-hydraulic properties and related studies, so that readers can better understand the importance of this research.

2. Clearly state the research hypothesis and objectives in the introduction, and further detail the research steps in the methodology section.

3. The discussion section should more clearly integrate and interpret the results, compare them with existing literature, and highlight the innovative points and significant findings of the research.

4. The description of the experimental design and sampling methods is brief. A more detailed description of each experimental step and control variables is needed to improve the reproducibility of the research.

5. Further explain the significance of each result in the discussion section, and compare them with existing literature to highlight the innovative points and limitations of this research.

6. Clearly state the limitations of the study (e.g., sample size, experimental conditions) in the discussion or conclusion section, and propose directions for future research improvements.

7. Improve the presentation of results by using clear charts and graphs, and provide detailed descriptions of each parameter and its units in the figure captions.

Comments on the Quality of English Language

Minor modifications required

Reviewer 2 Report

Comments and Suggestions for Authors

Processe of water retentionand movement are altered in rhizosphere. The aim of this study was to investigate the pchysical - hydric properties of soil agregates i the rhizosphere. The higher  activity observed in rhizosphere was faciliated by increased soil organic carbon and microbal activity and improved in soil agregation. The stability of agregates, increase in soil organic carbon. Increase of total porosity asocciated with soil aggregation. Soil pore space are influenced in higher Ksat. and hydraulik conductivity.

The abstract is short, concise and concise. The introduction well introduces the reader to the issues of changes in the physical properties of soils in the rhizosphere. The research results show different properties of soils in environments with different aggregate sizes. The drawings and charts are carefully made. Broad and correct discussion.

Why is the Materials and Methods chapter placed after the discussion chapter in the paper? The conclusions are accurate and concise.

References: Check Literature Line: 453, 464, 466, 492, 557, 564, 584, 589, 595, 597, 604, 609, 613, 616.

Reviewer 3 Report

Comments and Suggestions for Authors

Overall, the paper is well-written. The introduction is very comprehensive and the hypotheses are clear and well-described. Moreover the results are presented in a clear way and the discussion is comprehensive.

However, there are some methodological concerns. 1) the number of replicates (varying between 3 and 5 depending on the parameter) is extremely low for this type of research. It is rather surprising to have such low standard erors. Preferably, the number of replicates should be increased. 2) for some of the analyses, undisturbed soil samples should have been used as the disturbance of the soil has a major impact on the results (for example pore distribution and soil water retention curve). This is even mentioned by the authors in the discussion (lines 257-262). 3) although the authors mention the importance of the mucilage, it is desirable to include an assessment on active rootsin the experimental design.

Round 2

Reviewer 3 Report

Comments and Suggestions for Authors

The authors have made a well-reasoned argument for the various comments and remarks. I can definetly agree with their arguments.

They also stress themselves the limited nature of their research and the need for more advanced research techniques to conduct even deeper research.